# Assessment of the Safe Consumption of Nuts in Terms of the Content of Toxic Elements with Chemometric Analysis

**DOI:** 10.3390/nu13103606

**Published:** 2021-10-14

**Authors:** Joanna Bielecka, Anna Puścion-Jakubik, Renata Markiewicz-Żukowska, Jolanta Soroczyńska, Patryk Nowakowski, Monika Grabia, Konrad Mielcarek, Klaudia Przebierowska, Klaudia Kotowska, Katarzyna Socha

**Affiliations:** Department of Bromatology, Faculty of Pharmacy with the Division of Laboratory Medicine, Medical University of Białystok, Mickiewicza 2D Street, 15-222 Białystok, Poland; joanna.bielecka@umb.edu.pl (J.B.); renmar@poczta.onet.pl (R.M.-Ż.); jolanta.soroczynska@umb.edu.pl (J.S.); patryk.nowakowski@umb.edu.pl (P.N.); monika.grabia@umb.edu.pl (M.G.); konrad.mielcarek@umb.edu.pl (K.M.); klaudia130@gmail.com (K.P.); klaudiakkotowska@gmail.com (K.K.); katarzyna.socha@umb.edu.pl (K.S.)

**Keywords:** edible nuts, toxic elements, health risk assessment

## Abstract

Nuts are characterized by high nutritional value and are recommended as a part of a healthy diet. At the same time, toxic elements could also be found in them. In this research, we measured the content of As, Cd, Pb and Hg in a wide variety of edible nuts. To determine the As content, inductively coupled plasma mass spectrometry (ICP-MS) was applied. Cd and Pb were detected by the electrothermal atomic absorption spectrometry analytical technique (ETAAS) with Zeeman background correction, while atomic absorption spectrometry method (AAS) with the amalgamation technique in the case of Hg was used. The study material consisted of 120 samples without replications (10 for each subgroup) including the following nuts: Almonds, Brazil nuts, cashew nuts, hazelnuts, macadamia nuts, peanuts, pecan nuts, pine nuts, pistachios and walnuts. Indicators such as the target hazard quotient (THQ), cancer risk (CR) and hazard index (HI) were used to assess the health risk. The highest median As, Cd, Pb and Hg contents were observed for pistachios (192.42 µg/kg), pine nuts (238.40 µg/kg), peanuts (82.06 µg/kg) and pecans (82.06 µg/kg), respectively. The exceedance of the established limits was found in the case of Pb for nine samples: macadamia nuts (221.49 µg/kg; 2350.94 µg/kg; 2581.43 µg/kg), pine nuts (266.33 µg/kg), peanuts (1353.80 µg/kg) and pecans (2689.13 µg/kg, 2758.26 µg/kg, 2992.29 µg/kg and 3169.41 µg/kg). Extremely high (>2500 µg/kg) Pb content was found in 33% of studied pecans imported from the USA. The health risk indicators did not identify increased health risk. This research is significant considering the food safety issues and indicates the need to regularly control the content of toxic elements in food, as well as to establish the specific limits for heavy metals content in nuts. The chemometric analysis included cluster analysis and principal component analysis (PCA). Cluster analysis made it possible to distinguish four subgroups on the basis of the ability to accumulate toxic elements: pine nuts, pecans, pistachios and other analysed nuts. PCA indicated primarily factor 1, distinguishing mainly pecans, macadamia nuts and peanuts. Chemometric analysis can be a useful tool in estimating the ability of different nut species to accumulate contaminants.

## 1. Introduction

Nuts are characterized by high nutritional value and are recommended as a part of a healthy diet. Due to the high content of protein, mono- and polyunsaturated fats (MUFA and PUFA) and fibre as well as vitamins (folates, thiamine, vitamin E), minerals (magnesium, copper) and antioxidants, nuts can consist of a dietary source of these components. Taking into account their high caloric density, nuts intake was considered as a factor which could stimulate weight gain. However, current research does not support this thesis. It was demonstrated that nut intake could positively influence changes in metabolic factors such as glycaemic parameters or parameters of lipid metabolism. Moreover, the consumption of nuts was inversely associated with the occurrence of conditions such as hypertension, type 2 diabetes mellitus, obesity and cardiovascular diseases [1]. At the same time, some of the recent studies demonstrated that tree nuts could also be a source of exposure to toxic elements [2,3,4,5].

Arsenic (As) is most commonly transferred into the food chain from contaminated water or soil. Moreover, food alongside drinking water is regarded as the main route of human exposure to As [6]. As occurs in three major forms: organic, inorganic, and arsine gas (-3 oxidative state). Considering valence, there are three major states: As element (0), arsenite (+III) and arsenate (+V). Trivalent (+III) arsenic compounds (both organic and inorganic) are considered more toxic than pentavalent (+V) compounds. Inorganic As is regarded as being more toxic than the organic form [7]. As and inorganic As compounds, due to sufficient evidence, were classified as the first group of carcinogens for humans by the International Agency for Research on Cancer (IARC). It was observed that As exposure enhanced the risk of the occurrence of such cancers as skin, bladder, kidney, liver and lung cancer [8].

Cadmium (Cd) and its compounds, similarly to As, were classified as the first group of carcinogens for humans. Cd exposure is related to the development of several cancers—breast, stomach, kidney, urinary bladder, liver and pancreas. The mechanism involved in cancerogenesis is based on the induction of oxidative stress and irreversible damage to DNA. In non-smoking populations, Cd intake from food is a major source of exposure to this toxic element [8].

Lead (Pb) is the next toxic element which is frequently detected as a contaminant in various foodstuffs. Pb is highly toxic and could affect almost every organ in the human body. In children, even low levels of Pb exposure could result in behavioural disruption, learning problems and retarded growth. In adults, Pb could cause cardiovascular complications and impair the functioning of the kidneys and reproductive system. The IARC categorized Pb as probably cancerogenic to humans (group 2A) [8].

Mercury (Hg) is present in the environment in three forms: elemental (Hg), organic and inorganic forms. Elemental Hg is predominant form of Hg in atmosphere. Organic Hg is considered as the most toxic and accounts for the vast majority of Hg exposure. The most frequently detected organic forms are methylmercury (MeHg), and ethylmercury (EtHg). Inorganic forms obtain Hg salts of Hg^2+^ and Hg_2_^2+^; these compounds are present in disinfectants, fungicides and antiseptics. Each of those forms has different bioavailability and causes different toxic effects in the body. One of the most important causes of human exposure to Hg (in particular methylmercury) is the consumption of seafood. The critical organ affected by Hg toxicity is the brain; however, this element is also harmful to the heart, immune system, lungs and kidneys [9].

Long term exposure to even small amounts of toxic elements could be harmful to humans [10]. Diet is regarded as one of the most important sources of these compounds. Thereby, there is a great need to assess the toxicological aspects comprehensively and inform consumers not only about the nutritional aspects of nuts but also about the potential health risks. Due to specific climate conditions in Poland, only hazelnuts and walnuts are cultivated; the remaining nuts available on the market are imported from different parts of the world (North and South America, Africa, Asia and southern Europe). Numerous factors influence the mineral composition of nuts, among others natural environmental factors, climate characteristics, type of soil as well as transport and storage conditions [11]. Heavy metals could migrate to food from the packing materials [12]. Those elements are naturally present in the environment, e.g., as a result of soil erosion. Environmental pollution varies across the continents. In contaminated areas, toxic elements are transferred through the air, water and soil to plants and consequently to the food chain.

The objective of this study was to determine the content of toxic elements (As, Cd, Pb and Hg) in ten of the most popular types of edible nuts and to assess the health risks resulting from their consumption. Furthermore, the differences in the content of the studied elements regarding the type of nut as well as the country of origin were determined and discriminated by multivariate analysis.

## 2. Materials and Methods

### 2.1. Reagents

The reagents purchased from the Merck Company (Darmstadt, Germany) were: ultrapure concentrated nitric acid (69%), standard solutions of studied elements—arsenic (H_3_AsO_4_ in HNO_3_ 0.5 mol L^−1^, 1000 mg L^−1^ As), cadmium (Cd(NO_3_)_2_ in HNO_3_ 0.5 mol L^−1^, 1000 mg L^−1^ Cd), lead (Pb(NO_3_)_2_ in HNO_3_ 0.5 mol L^−1^, 1000 mg L^−1^ Pb), mercury (Hg(NO_3_)_2_ in HNO_3_ 2 mol L^−1^, 1000 mg L^−1^ Hg). From Sigma-Aldrich (Saint Louis, MO, USA) ammonium dihydrogen phosphate was obtained. Certified reference material mixed polish herbs (INCT-MPH-2) was obtained from the Institute of Nuclear Chemistry and Technology (Warsaw, Poland).

### 2.2. Sample Collection

The study material consisted of ten types of nuts: Almonds (*Prunus dulcis* L.), Brazil nuts (*Bertholletia excels* L.), cashew nuts (*Anacardium occidentale* L.), hazelnuts (*Corylus avellana* L.), macadamia nuts (*Macadamia ternifolia* F.), peanuts (*Arachis hypogaea* L., which botanically belongs to legume family), pecan nuts (*Carya illinoinensis* L.), pine nuts (*Pinus pinea* L.), pistachios (*Pistacia vera* L.) and walnuts (*Juglans regia* L.). A total of 120 samples were purchased from markets in standard packages, as well as by weight, between January and March 2021. Ten samples (each of the different producers) were obtained among every studied subgroup without replications. Then, the nuts were transferred into non-sterile polytetrafluoroethylene containers and were stored in a shaded place at room temperature until analyses.

### 2.3. Sample Preparation and Digestion

To obtain a uniform texture of samples for the mineralization process, samples were homogenized using T 18 digital Ultra-Turrax (IKA, Staufen, Germany). Then, an amount of between 200 and 300 mg was weighted to polytetrafluoroethylene vessels and 4 mL spectrally pure concentrated (69%) HNO_3_ was added (Merck, Darmstadt, Germany). The microwave mineralization was carried out in a close-loop system, applying the procedure described previously (Speedwave, Berghof, Germany) [13]. The digestion procedure consisted of 3 steps. The temperature, pressure, time and power of the microwave generator in the following steps were: 170/190/210 °C, 20/30/40 atm, 10/10/10 min and 80/90/90%, respectively. The final step was to cool the samples at temp. 50 °C, pressure 40 atm, time 18 min and 0% of power. The closed digestion system ensures clean sample preparation and no loss of analyte in the case of volatile elements such as As. Digested samples were quantitatively transferred with ultrapure water (Millipore Simplicity UV Water Purification System, Merck, Darmstadt, Germany) to polypropylene vessels and stored at −20 °C before determination of As, Cd and Pb. After all mass of mineralized samples ranged from 4.264 g to 6.002 g. Hg content was determined without digestion. 

### 2.4. Analysis of Toxic Elements Content in Nuts

#### 2.4.1. Arsenic

Inductively coupled plasma-mass spectrometry (ICP-MS, NexION 300D, Perkin Elmer, Waltham, MA, USA) with a kinetic energy discrimination (KED) chamber was used for As measurement. In this configuration, collisions and kinetic energy discriminations are used to correct polyatomic interferences. The conditions on the basis of which the measurements were carried out were: mass—75 amu, dwell time per amu—50, integration time—1000 ms, and dual calibration mode. The results were obtained as a count per second and were converted into concentrations based on calibration curves. The limit of As detection (LOD) was 0.019 µg/kg. For As measurement mineralized samples were diluted 10 times with ultrapure water.

#### 2.4.2. Cadmium and Lead

Electrothermal atomic absorption spectrometry analytical technique (ETAAS) with Zeeman background correction for Cd and Pb measurement was used (Z-2000, Hitachi, Japan). We used pyrolytically coated graphite cuvettes with widened central section (PyroTube CII HR, Hitachi), particularly suitable for high-sensitivity analyses. The determination of Cd and Pb was performed at the wavelengths of 228.8 nm and 283.3 nm, respectively. We used 0.5% ammonium dihydrogen phosphate (NH_4_H_2_PO_4_, Sigma-Aldrich, Saint Louis, MO, USA) as a matrix modifier. The limit of detection value for Pb and Cd determination was 1.24 µg/kg and 0.02 µg/kg, respectively.

#### 2.4.3. Mercury

The atomic absorption spectrometry method (AAS) with the amalgamation technique in a single-purpose atomic absorption spectrometer for Hg determination was applied (AMA-254, Leco Corp, Altec Ltd., Prague, Czech Republic). Appropriate amounts of samples (about 50 mg with accuracy 0.1 mg) were placed in the cuvette and analysed. The measurement procedure consists of three steps. First, the samples were burned at 600 °C in oxygen. Then, the vapours of Hg passed through the catalytic column to the amalgamator. Additionally, in the last phase after release from the amalgamator, Hg was measured at a wavelength of 254 nm. The steps lasted 60 s, 150 s and 45 s, respectively. The limit of detection was 0.003 ng/sample.

### 2.5. Quality Control of the Analytical Methods Used

Quality control was carried out prior to the analyses and every 10 samples through analysing certified reference material (mixed Polish herbs (INCT-MPH-2) (Institute of Nuclear Chemistry and Technology, Warsaw, Poland). Obtained results were referred to the standard values provided by the manufacturer. The recovery for As, Cd, Pb and Hg were: 103%, 105%, 98.5%, and 102.5%, respectively, while the precision was as follows: 2.1%, 2.6%, 3.1%, 2.9%.

### 2.6. Health Risk Assessment

To assess short and long-term adverse effects (taking into account cancerogenic and non-cancerogenic effects) due to exposure to studied toxic elements following indicators were calculated: estimated daily intake (EDI), cancer risk (CR), target hazard quotient (THQ) and hazard index (HI). Health risk indicators were calculated based on formulas described previously:EDI = (C × Cons)
CR = (Fr × D × EDI × Sf)/T × 10^−3^
THQ = (Fr × D × Cons × C)/(RfD × BW × T) × 10^−3^
HI = ∑(THQ_As_ + THQ_Cd_ + THQ_Pb_ + THQ_Hg_),
where C is the concentration of studied element in sample, Cons is the average level of consumption, Fr is the frequency of exposure (365 days/year), D is the duration of exposure (in this work we have taken the average lifetime of 70 years), Sf is slope factor established by United States Environmental Protection Agency for As—1.5 mg/kg/day, for Cd—6.3 mg/kg/day, and for Pb—0.0085 mg/kg/day, T is the overall time of exposure (Fr × D), RfD is oral reference dose for As 0.3 µg/kg BW/day, for Cd and Pb 1 µg/kg BW/day, and for Hg 0.3 µg/kg BW/day and BW is the average body weight [14].

The average daily consumption was estimated at 42 g [15], while an average body weight of 70 kg was taken. The European Commission regulations do not specify the norms for As, Cd, Pb and Hg in nuts, so we used the Polish National Food Safety Standard to compare our results with the established levels. The maximum Cd and Pb in nuts were founded to be 500 µg/kg and 200 µg/kg, respectively, while the levels of As and Hg in nuts were not included in this regulation [16].

### 2.7. Statistical and Multivariate Analysis

The obtained results were analysed using Statistica 13 software (TIBCO Software Inc., Palo Alto, CA, USA). To assess the normality of data two tests were performed—Shapiro–Wilk and Kolmogorov–Smirnov. No criterion of the normality was observed, thus, Kruskal–Wallis Analysis of Variance (ANOVA) was used to compare the content of toxic elements among the studied products. Taking into account the country of origin, 81 samples were analysed; the remaining 39 products lacked this information. Most of the nuts were imported from Asia (35), followed by 15 from the North America, 11 from European countries and 11 from Australia, as well as 9 from South America. Significant differences were considered on the levels of *p* < 0.05, *p* < 0.01 and *p* < 0.001. The results were described as the median and interquartile range (quartile 1–quartile 3, Q_1_–Q_3_). However, to make comparisons with other authors easier, the mean and minimum–maximum range were also added. A chemometric analysis was also performed, which included cluster analysis, principal components analysis (PCA) and correspondence analysis.

## 3. Results and Discussion

### 3.1. Content of Toxic Elements in Nuts

In our study, the content of toxic elements varied considerably between the types of nut samples (Table 1). We have observed that pecans were generally characterized by a high content of each of the elements studied. All pecan samples were imported from the USA; however, there was no detailed information about the specific area of origin. To the best of our knowledge, there is no research which could explain whether pecans or other nuts accumulate greater amounts of toxic elements. Most probably, the contamination of nuts by toxic elements is the result of environmental pollution [17]. Foodstuffs could also be contaminated with toxic elements through migration from food packages. The levels of contamination vary between selected elements and different packaging materials [18]. Generally, among the studied nuts, we can classify a group of nuts that accumulate toxic elements to a lesser extent, and that group includes almonds, Brazil nuts, cashews, hazelnuts, and walnuts. On the other hand, in the group of nuts which was characterized by increased contents of studied elements were: macadamia nuts, peanuts, pecans, pine nuts and pistachios.

#### 3.1.1. Arsenic

In our study, the highest median As content (Table 1) was detected among pistachio nuts (192.42 µg/kg) and the lowest in the subgroup of almonds (23.59 µg/kg). The As levels in the studied material ranged from 13.48 µg/kg in one sample of the cashews to 314.52 µg/kg in the sample of pecans. Those pecans were imported from the USA. The Polish National Food Safety Standard does not determine the maximum levels of As in nuts, therefore, we could not refer our results to any standard. Among the results of the other authors, the highest median As was reported among walnuts (200 µg/kg—which was four times higher than in this research) as well as results similar to ours in pistachios (200 µg/kg) [19]. According to the literature, the lowest As concentrations were determined to be found in the samples of Brazil nuts [2,20,21,22]. As its uptake by plants is dependent on its total concentration (considering speciation) in the soil and on the bioavailability, As is most effectively absorbed with protein transporters in inorganic forms—As(III) and As(V). As could alter the morphology and biochemistry and could cause changes at the molecular level, which consequently affect plants’ growth and productivity [23]. Among the other foods that could accumulate As in significant amounts, the following products were included: seafood, fruits and vegetables, rice and drinking water. The detected As amounts ranged from 2 to 932 µg/kg [24]. Moreda-Piñeiro et al. in their study, assessed the bioavailability of essential and toxic metals in nuts. As content in hazelnuts and pistachios was 78.9 µg/kg and 177 µg/kg, while the dialyzable As levels were 59.4 µg/kg and 90.7 µg/kg, respectively. That indicates a high percentage of bioavailability of this element from those types of nuts (75.2% and 51.1%). In the remaining study material, the bioavailability was not determined due to the As contents being lower than the LOD (0.019 µg/kg) [3].

#### 3.1.2. Cadmium

Considering Cd content, the highest median (238.40 µg/kg) was found for the tested pine nuts (Table 1). On the other hand, the lowest median (0.54 µg/kg) was observed among macadamia nuts, while, in the individual samples, Cd concentration was between 0.09 µg/kg and 458.22 µg/kg for one sample of macadamia nuts and pine nuts, respectively. The sample of pine nuts with the greatest Cd content was imported from China. Taking into account the results of other studies, the highest mean Cd level (695 µg/kg) was determined for hazelnuts, and the lowest (<2 µg/kg) for both almonds and walnuts [4,22]. In our research, the exceedance of the limit (500 µg/kg) of the Polish National Food Safety Standard (PNFSS) was not observed; however, in eight samples Cd levels were between 200 and 500 µg/kg [16]. Cd is absorbed into plants from the soil and transferred to the fruits. Industrial and urban emissions are listed as being among the main sources of Cd contamination of the soil. Cd concentrations in food depend on the geographical localization, the bioavailability from the soil, as well as plants’ genetics and the fertilizers used [25]. In non-smoking populations, diet is a major source of Cd exposure [26]. The food products which have a higher cadmium accumulation potential are crustaceans, offal (liver, kidneys), nuts, vegetables, coffee, tea and cocoa. The Cd levels in those products ranged from 100 to 4800 µg/kg [27]. It was observed that Cd accumulation in almond seedlings increased with the external concentrations of this toxic element. Moreover, Cd stress was related to changes in total fatty acids content in all classes [28]. The in vitro bioaccessibility of Cd from hazelnuts and walnuts was between 26 and 27% in the gastric juice and between 42 and 45% in the intestinal juice [29]. However, a lower bioavailability of Cd in nuts was determined by Moreda-Piñeiro et al. The authors obtained the following results for macadamia nuts (4.4%), pecans (2.2%), hazelnuts (3.5%), peanuts (2.2%), pine nuts (2.3%) and pistachios (1.4%) [3].

#### 3.1.3. Lead

The analysis of the Pb content in the nuts (Table 1) showed that the highest median (82.06 µg/kg) was reached by peanuts, while the highest mean concentration of Pb (980.37 µg/kg) was found in pecan nuts. The lowest median (10.95 µg/kg) and mean (13.08 µg/kg) concentrations of Pb were in walnuts. The scatter of Pb concentration in tested nuts was from 1.71 µg/kg in macadamia to 3169.41 µg/kg in pecans. The sample of pecans found to have the highest Pb level was imported from the USA. Comparing our findings with the results of other authors, it was found that the Pb content in Brazil nuts, macadamia nuts and pecans was higher than in all the other scientists’ analyses [2,3,4,20,22,30]. In our study, the Pb levels in one sample of peanuts (1353.80 µg/kg), two (17%) samples of macadamia nuts (2350.94 and 2581.43 µg/kg) and four (33%) of pecans (2689.13, 2758.26, 2992.29 and 3169.41 µg/kg) considerably exceeded the Polish National Food Safety Standard established level of 200 µg/kg. Besides, two samples—one of macadamia nuts (221.49 µg/kg) as well as one of pine nuts (266.33 µg/kg)—had higher Pb contents than the upper limit [16]. Among the products that could substantially contribute to Pb exposure are cereal products and grains, vegetables, tea and milk products. The Pb concentration ranges varied in those products from 0.3 µg/kg to 4300 µg/kg [31].

#### 3.1.4. Mercury

During the analysis of Hg concentration in nut samples (Table 1), it was shown that pecans nuts reached the highest median (5.77 µg/kg) and the highest mean concentration (11.64 µg/kg) of Hg. On the other hand, the lowest median (1.67 µg/kg) and the lowest average concentration (2.40 µg/kg) of Hg were detected in Brazil and macadamia, respectively. The range of the determined values of Hg concentration in all samples was from 0.23 µg/kg in pine nuts to 44.74 µg/kg in walnuts. Those walnuts were grown in Poland. The results were comparable with those obtained by other authors. The only exceptions were pecan and pine nuts, which showed results different from those published in the literature [20,32]. Taking into account the bioaccessibility of toxic elements from nuts, Moreda-Piñeiro et al. found that relatively low amounts of Hg were dialyzed from hazelnuts. Hg in raw hazelnuts was 104.6 µg/kg, of which 0.96 µg/kg was dialyzed [3]. Among the other foodstuff that contains high amounts of Hg, seafood and fish were found. However, research conducted in recent years has indicated that rice, especially when consumed as a staple food, could also pose a source of exposure [33].

**Table 1 nutrients-13-03606-t001:** Content of toxic elements in nuts in this study and determined by other authors.

Nuts	As (µg/kg)		Other Authors	Cd (µg/kg)		Other Authors	Pb (µg/kg)		Other Authors	Hg (µg/kg)		Other Authors
	This Study			This Study			This Study			This Study		
	X ± SD Min-Max	Me Q_1_–Q_3_	X ± SD	X ± SD Min–Max	Me Q_1_–Q_3_	X ± SD	X ± SD Min–Max	Me Q_1_–Q_3_	X ± SD	X ± SD Min–Max	Me Q_1_–Q_3_	X ± SD
Almonds	23.15 ± 2.34 18.51–26.88	23.59 21.84–24.45	74.77 ± 13.04 [4] 12 ± 1 [3] 6.6 ± 4.0 [33]	15.75 ± 12.70 0.89–51.65	14.7410.44–16.47	350 ± 80 [22] 480 ± 240 [21] 36.85 ± 4.47 [4] <2 [3]	15.06 ± 10.60 5.21–44.01	12.80 8.05–18.53	70 ± 2 [22] 281 ± 56 [21] 57.92 ± 28.48 [4] <6 [3]	5.84 ± 6.00 1.45–23.66	4.25 3.50–5.04	1.4 ± 0.3 [30] <7 [3] 0.72 ± 0.45 [4] 56 ± 6 [32]
Brazil Nuts	27.12 ± 8.53 21.17–45.36	24.32 22.74–25.17	50.34 ± 2.61 [4] 9 ± 1 [3] <3 [34] 1.7 ± 0.97 [4]	1.59 ± 1.43 0.36–5.15	1.29 0.44–1.84	1.85 ± 0.25 [4] 8 ± 1 [3] <5 [34] <31 [35]	82.11 ± 12.85 63.03–107.02	82.47 76.38–88.32	12.16 ± 0.05 [4] <6 [3] <19.3 [36] 1.7 ± 0.97 [4]	2.97 ± 3.85 0.97–14.60	1.67 1.22–2.33	1.3 ± 0.4 [30] <7 [3] <2 [34] 0.12 ± 0.05 [4]
Cashews	33.64 ± 18.31 13.48–71.46	26.50 24.77–31.96	44.66 ± 3.46 [4] 15 ± 1 [3] 5.3 ± 4.7 [4] Me = 150 [2]	13.25 ± 28.51 1.05–99.14	2.46 1.27–5.44	400 ± 210 [21] 0.99 ± 0.46 [4] 12 ± 1 [3] 0.8 ± 0.97 [4]	20.05 ± 18.12 2.44–53.99	12.06 7.89–30.94	661 ± 68 [21] 104.5 ± 19.5 [4] <6 [3] 2.4 ± 1.4 [4]	4.54 ± 3.36 1.12–11.22	3.89 1.56–7.26	1.6 ± 0.5 [30] <7 [22] 0.78 ± 0.46 [4] 69 ± 7 [32]
Hazelnuts	29.49 ± 14.69 21.34–72.17	23.96 23.54–26.11	78.9 ± 5.4 [37] 48.94 ± 3.37 [4] 24 ± 1 [3] 11 ± 15 [33]	11.78 ± 6.93 3.43–23.22	11.00 5.40–16.33	695 ± 27 [22] 5.5 ± 0.47 [37] 58.81 ± 4.95 [4] 22 ± 0.1 [3] 10 ± 1.6 [4]	75.77 ± 9.42 64.52–94.91	74.04 68.66–81.06	138 ± 20 [22] 99.47 ± 10.47 [4] <6 [3] 7.3 ± 8.2 [4]	4.53 ± 4.69 1.37–18.38	2.77 2.30–4.04	104.6 ± 2.1 [37] 32 ± 1 [3] 2.2 ± 0.5 [4]
Macadamia Nuts	38.47 ± 32.70 19.78–121.29	23.49 22.67–30.61	Me = 180 [2]	0.69 ± 0.57 0.09–1.87	0.54 0.42–0.66	481 ± 15 [4] 5.2 ± 0.03 [37] 460 ± 230 [21] Me = 224 [2]	437.07 ± 951.00 1.71–2581.43	11.44 8.94–67.54	84 ± 8 [22] Me = 267 [2]	2.40 ± 1.41 1.07–5.47	1.92 1.75–2.44	-
Peanuts	36.07 ± 18.98 21.10–71.20	23.95 22.03–46.14	48.49 ± 4.27 [4]	84.49 ± 69.32 38.58–292.23	62.68 48.99–82.19	5.5 ± 0.09 [37] 520 ± 190 [21] 36.84 ± 3.51 [4] Me = 610 [2]	188.55 ± 367.15 66.76–1353.80	82.06 74.55–93.57	1862 ± 225 [21] 24.30 ± 3.41 [4] Me = 131 [2] 160 [17]	2.71 ± 1.06 1.11–4.57	2.46 2.08–3.59	2.0 ± 0.2 [30] 6 [17] 119 ± 14 [32]
Pecans	144.12 ± 106.68 23.60–314.52	150.27 39.23–221.08	4.5 ± 1.4 [4] 150 [20]	74.50 ± 35.69 27.11–123.02	84.10 33.06–100.72	3.5 ± 0.55 [37] 45 ± 26 [33] 180 [20]	980.37 ± 1424.09 12.02–3169.41	21.23 16.95–2706.41	0.82 ± 0.39 [4] 5.9 [20]	11.64 ± 13.48 2.04–41.72	5.77 2.55–14.19	2.8 ± 1.1 [4]
Pine Nuts	53.78 ± 37.46 22.13–141.32	47.072 2.71–73.77	Me = 200 [2] 160 [20]	246.87 ± 172.28 35.72–458.22	238.40 90.85–408.99	5.1 ± 0.7 [37] 490 ± 320 [21] Me = 380 [2] 110 [20]	48.82 ± 70.71 11.93–266.33	23.73 18.43–39.48	730 ± 250 [21] Me = 121 [2] 21 [20]	3.97 ± 1.97 0.23–6.68	4.23 3.12–5.19	78 ± 8 [32]
Pistachios	188.80 ± 54.38 92.84–280.90	192.42 153.96–216.20	177 ± 5.3 [37] 66.56 ± 0.71 [4] Me = 200 [2]	4.48 ± 3.88 0.62–13.27	2.72 2.09–5.62	264 ± 3 [22] 3.2 ± 0.49 [37] 450 ± 230 [21]	21.73 ± 20.34 3.25–77.88	15.80 10.16–27.52	42 ± 4 [22] 1162 ± 142 [21] 20.92 ± 5.30 [4] Me = 118 [2]	3.82 ± 2.82 1.31–9.94	2.83 2.14–3.76	1.8 ± 0.4 [30] 32 ± 4 [32]
Walnuts	41.08 ± 12.99 19.27–64.57	45.19 31.99–46.91	64.01 ± 3.41 [4] 19 ± 1 [3] 7.1 ± 6.1 [4] Me = 200 [2]	5.48 ± 10.04 0.15–33.64	1.80 0.86–2.54	385 ± 19 [22] 20 ± 30 [19] 490 ± 320 [21]<2 [3]	13.08 ± 6.01 5.70–22.26	10.95 8.90–18.40	64 ± 3 [22] 100 ± 170 [19] 9.26 ± 2.19 [4] <6 [3]	7.52 ± 12.81 0.94–44.74	1.94.65–4.60	1.75 ± 0.9 [19] 1.8 ± 0.2 [30] 15 ± 1 [3] 0.8 ± 0.33 [4]

Max—maximum; Me—median; Min—minimum; Q_1_—lower quartile; Q_3_—upper quartile; SD—standard deviation; X—mean.

### 3.2. Significance Assessment, Correlation and Chemometric Analysis

Several differences in the content of the studied toxic elements were determined between the subgroups of nuts (Table 2). The *p*-values were put in the superscript.

The correlation coefficient determines the relationship between two parameters and ranges from +1 to −1. Our analyses showed a strong, negative correlation between the content of As and Hg in Brazil nuts (r = −0.86, *p* < 0.001), between Pb and Hg in hazelnuts (−0.60, *p* < 0.05) and in pecan between the content of As and Cd (−0.73, *p* < 0.01). Moreover, we showed a strong positive correlation between the Cd and Pb content in cashews (r = 0.80, *p* < 0.01). Other correlations are presented in Figure 1. These correlations may indicate a tendency to accumulate nuts’ individual contaminants—this indicates similar plant uptake rates when using different plant channels.

Cluster analysis aims to group objects in terms of their similarity—in this case, the median content of the examined elements was used. The Euclidean distance was used as a measure of similarity, and clustering was performed by Ward’s method. Four main clusters were obtained: the first containing pine nuts, the second—pecan, the third—pistachio, and the fourth—the remaining types of nuts (Figure 2). With reference to the above results, pine nuts had the highest median Cd content, pecan nuts—the highest Pb and Hg content, and pistachio—the highest median As content. Cluster analysis allowed us to distinguish nut species with the highest medians of the tested toxic elements.

PCA is a method which, by transforming the output variables into new variables, builds a model describing the relationships between them. In the case of the content of toxic elements in nuts, a specific model was obtained, because the first component explains as much as 96.55% of the total variance. According to the Cattell scree plot, only 1 component should be classified for interpretation—in order to create a two-dimensional plot, the first two components were classified, explaining a total of 98.83% of the variance. The variable Pb has a coefficient of −0.9997 for the first factor, and the variable Cd −0.9993 for the second factor. Factor 1 distinguishes, among others, pecan nuts, macadamia nuts, and peanuts (Figure 3).

Correspondence analysis allows us to obtain information about the relationships between the categories of variables. When assessing the relationship between the origin and type of nuts, for the first two dimensions, the cumulative percentage of the eigenvalue is 56.17% (Figure 4).

Chemometric analysis, based on the content of selected elements, including As, Cd and Pb, was performed by Kafaoğlu et al. [2]. Our analysis showed a low positive correlation between the content of Cd and Pb in all nuts in general (r = 0.26, *p* < 0.01). The cited authors showed a similarly low positive correlation (r = 0.18). Moreover, the authors showed a similarity in the content of the elements, despite their different origins. Our analysis allowed us to identify the quality features of the nuts.

### 3.3. Health Risk Assessment

The health risk from intake of the specific species of nuts was calculated based on EDI, THQ, HI and CD indicators. The mean values of health risk indicators of toxic elements from tested nuts, such as EDI, CR, THQ and HI, were presented in Table 3. These indicators are valuable parameters for assessing the health risk of intoxication with the toxic metals associated with the intake of specific nuts. Human consumption of products which exceed the admissible content of toxic elements may have serious health consequences [35]. The lowest EDI, CR and THQ for Cd values were found in smoked macadamia nuts, and they reached 2.91 × 10^−5^, 1.83 × 10^−7^ and 4.16 × 10^−4^, respectively. The lowest indicator values were 5.49 × 10^−4^, 4.67 × 10^−9^ and 2.24 × 10^−34^ in walnuts for Pb; 9.73 × 10^−4^, 1.46 × 10^−6^ and 4.63 × 10^−2^ in almonds for As; 1.01 × 10^−4^ and 4.80 × 10^−3^ in macadamia nuts for Hg, respectively.

The highest EDI, CR and THQ calculated for Cd were found in pine nuts (EDI = 1.04 × 10^−2^, CR = 6.53 × 10^−5^, THQ = 1.48 × 10^1^), for Pb in pecans (EDI = 4.12 × 10^−2^, CR = 3.50 × 10^−7^, THQ = 1.68 × 10^−1^), for As in pistachio nuts (EDI = 7.93 × 10^−3^, CR = 1.19 × 10^−3^, THQ = 3.78 × 10^−1^) and for Hg in pecans (EDI = 4.89 × 10^−4^, CR = 2.33 × 10^−2^, THQ = 5.24 × 10^−1^). It has been found that consumption of studied species of nuts does not cause a carcinogenic risk via the content of toxic elements. None of the tested samples exceed the acceptable value of THQ and their consumption is safe for human health. The totality health risk of all toxic elements (HI) found in the samples of selected nuts species was the lowest in almonds (7.00 × 10^−2^) and the highest in pecans (5.24 × 10^1^). The value of HI is potentially dangerous when it exceeds 1; thus, it can be concluded that the consumption of a standard portion of all studied nut species is safe for human health. Summarizing the obtained results of the health risk assessment, it was found that the consumption of a standard portion of nuts daily does not pose a threat to human health.

After a detailed analysis of the available literature, it was shown that the current knowledge was lacking information about the assessment of health risks resulting from the consumption of portions of nuts contaminated with toxic elements. Currently, only publications on the health risk assessments of walnuts and pistachios were available. In the case of walnuts, the THQ values obtained in the research for Cd, Pb and As were lower than those obtained by other authors (Cd: 1.00 × 10^−1^, Pb: 9.00 × 10^−2^ and As: 2.00 × 10^−2^). On the other hand, the THQ value for Hg in the tested walnuts was higher than the results available in the literature (Hg: 8.00 × 10^−2^) [38]. Taghizadeh et al. determined risk of endpoints posed through exposure to toxic elements, such as As and Pb, via the consumption of walnuts from Iran in the general population. Based on his research, the THQ calculated for As were 2.30 × 10^−1^ and 0.04 × 10^1^ for the 50th and 95th centiles and for Pb was 1.03 × 10^−3^ and 2.01 × 10^−3^ for the 50th and 95th centiles, respectively. The CR value of walnuts for As and Pb reached 1.03 × 10^−4^ and 3.11 × 10^−4^ for As and 4.71 × 10^−9^ and 1.05 × 10^−8^ for Pb for the 50th and 95th centiles, respectively [39]. The health risk assessment of exposure to As, Cd and Pb after intake of walnuts was also examined by Wu et al. and reached values of 0.02 × 10^0^, 0.07 × 10^−1^ and 0.01 × 10^0^, respectively. These walnuts cultivated in China reached THQ values higher than the results obtained in our research [40]. Pistachios were another species of nuts whose health risk assessment was carried out. The health risk assessment resulting from the As content in the nuts showed that the THQ value of the tested pistachios was higher than the value of Iranian pistachios, available in published scientific studies (As—4.50 × 10^−2^). The HI calculated in our research was lower than the value determined by Taghizadeh et al. (HI—4.68 × 10^−1^) [39].

## 4. Conclusions

In our study, the content of toxic elements varied considerably between types of nut samples, which was confirmed by chemometric analysis. To our knowledge, this is the first study that comprehensively evaluated the health risks due to the consumption of nuts among the Polish population. The exceedance of the established limit of Pb was found in nine samples (7.5%), while 33% of tested pecans had extremely high (over 2500 µg/kg) Pb levels. However, the other tested products seem to be safe to consume. Not only the content of the toxic elements in food products, but also the level of their consumption plays a crucial role in evaluating the health risk. Furthermore, we see a great need to establish the maximum levels of toxic elements in nuts in the official European regulations. Foods available on the market should be controlled regularly, since food safety and quality are included in the most important public health issues.

## Figures and Tables

**Figure 1 nutrients-13-03606-f001:**
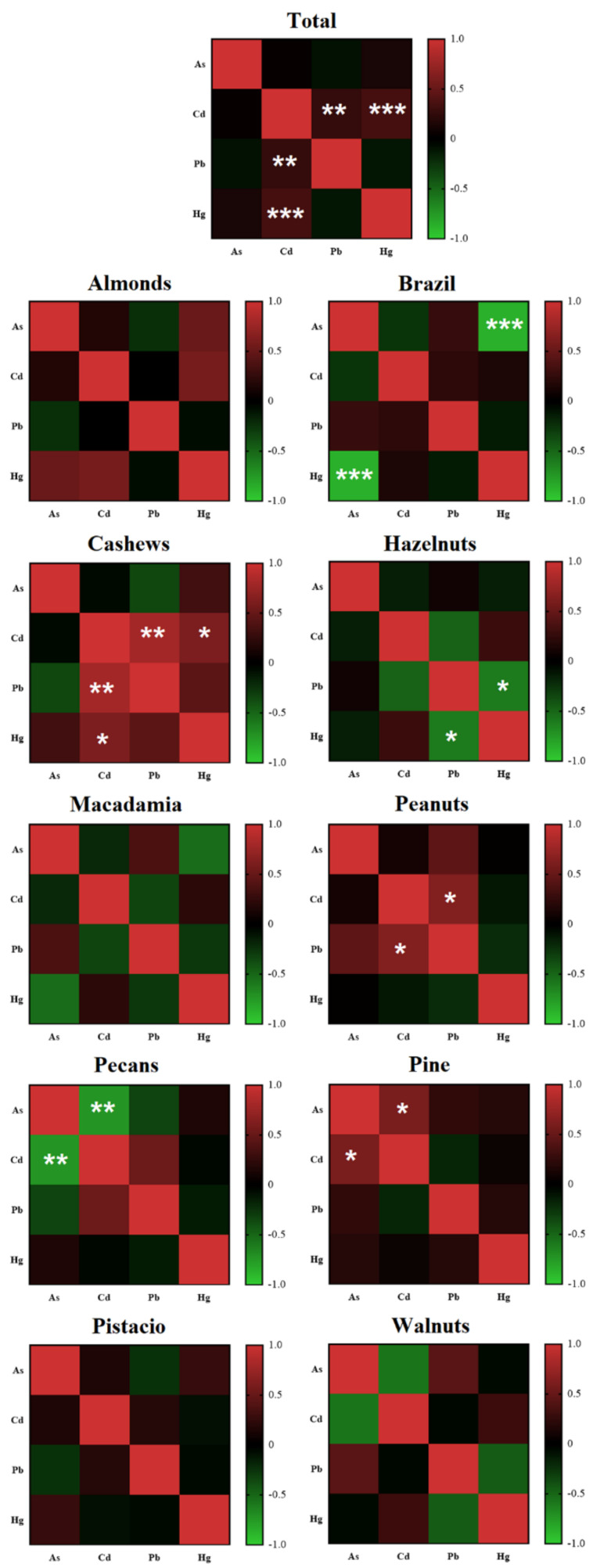
A heatmap of the Spearman correlation between the content of toxic elements. A—almonds; B—Brazil nuts; C—cashews; H—hazelnuts; M—macadamia; Pea—peanuts; Pec—pecans; Pin—pine; Pis—pistachio; W—walnuts, *****
*p* <0.05, ******
*p* <0.01, *******
*p* <0.001.

**Figure 2 nutrients-13-03606-f002:**
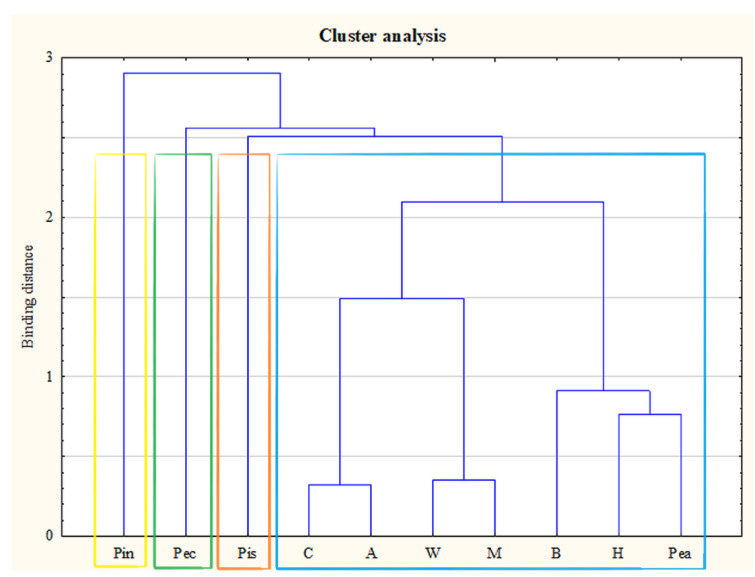
Dendrogram based on the content of toxic elements. A—almonds; B—Brazil nuts; C—cashews; H—hazelnuts; M—macadamia nuts; Pea—peanuts; Pec—pecans; Pin—pine; Pis—pistachios; W—walnuts.

**Figure 3 nutrients-13-03606-f003:**
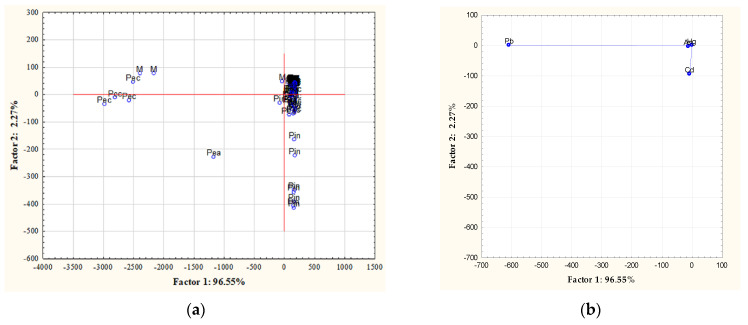
Two—dimensional plot of factor coordinates for cases (**a**) and variables (**b**). A—almonds; B—Brazil nuts; C—cashews; H—hazelnuts; M—macadamia; Pea—peanuts; Pec—pecans; Pin—pine; Pis—pistachio; W—walnuts.

**Figure 4 nutrients-13-03606-f004:**
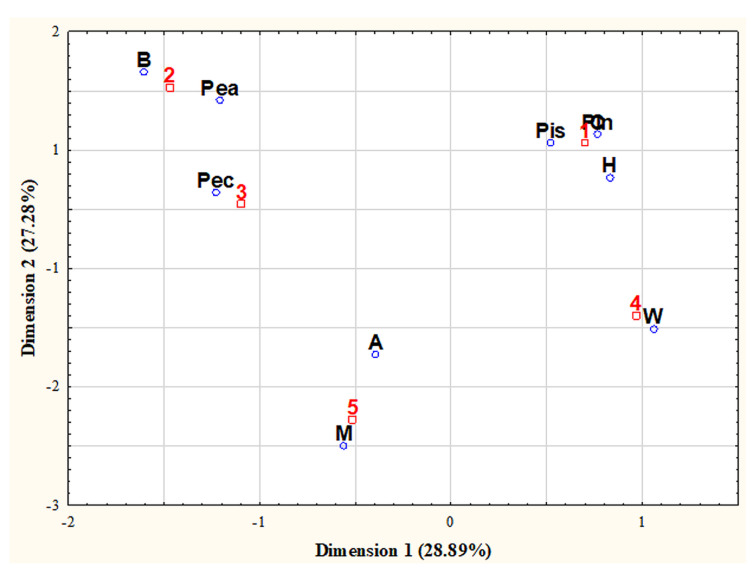
Two—dimensional graph for the tested nuts. A—almonds; B—Brazil nuts; C—cashews; H—hazelnuts; M—macadamia; Pea—peanuts; Pec—pecans; Pin—pine; Pis—pistachios; W—walnuts; 1—Asia; 2—South America; 3—North America; 4—Europe; 5—Australia.

**Table 2 nutrients-13-03606-t002:** Statistically significant differences in the content of the studied elements between tested nuts.

	Almonds	Brazil Nuts	Cashews	Macadamia Nuts	Pecans	Pine Nuts	Pistachios	Walnuts
Brazil Nuts	Pb^0.001^		Pb^0.001^	Pb^0.05^	Cd^0.001^ As^0.05^ Hg^0.05^	Cd^0.001^	Pb^0.001^ As^0.05^	Pb^0.001^
Cashews					Cd^0.05^	Cd^0.001^	As^0.05^	
Hazelnuts	Pb^0.001^		Pb^0.001^	Cd^0.05^			Pb^0.05^ As^0.001^	Pb^0.05^
Macadamia Nuts	Cd^0.01^				Cd^0.001^	Cd^0.001^	As^0.001^	
Peanuts	Pb^0.05^	Cd^0.001^	Cd^0.05^ Pb^0.001^	Cd^0.001^ Pb^0.05^			Cd^0.05^ Pb^0.001^ As^0.001^	Cd^0.001^ Pb^0.001^
Pecans	As^0.001^	As^0.05^					Cd^0.05^	Cd^0.001^
Pine Nuts							Cd^0.001^	Cd^0.001^
Pistachios	As^0.001^							

**Table 3 nutrients-13-03606-t003:** Estimated human health risk calculated for toxic elements (Hg, Pb, Cd and As) through consumption of selected species of nuts.

Nuts	As			Cd			Pb			Hg		HI
	EDI (mg/Day)	CR	THQ	EDI (mg/Day)	CR	THQ	EDI (mg/Day)	CR	THQ	EDI (mg/Day)	THQ	
Almonds	9.73 × 10^−4^ ± 9.82 × 10^−5^ (7.77 × 10^−4^–1.13 × 10^−3^)	1.46 × 10^−6^ ± 1.47 × 10^−7^ (1.17 × 10^−6^–1.69 × 10^−6^)	4.63 × 10^−2^ ± 4.68 × 10^−3^ (3.70 × 10^−2^–5.38 × 10^−2^)	6.61 × 10^−4^ ± 5.33 × 10^−4^ (3.72 × 10^−5^–2.17 × 10^−3^)	4.17 × 10^−6^ ± 3.36 × 10^−6^ (2.34 × 10^−7^–1.37 × 10^−5^)	9.45 × 10^−3^ ± 7.62 × 10^−3^ (5.31 × 10^−4^–3.10 × 10^−2^)	6.32 × 10^−4^ ± 4.45 × 10^−4^ (2.19 × 10^−4^–1.85 × 10^−3^)	5.38 × 10^−9^ ± 3.78 × 10^−9^ (1.86 × 10^−9^–1.57 × 10^−8^)	2.58 × 10^−3^ ± 1.82 × 10^−3^ (8.92 × 10^−4^–7.54 × 10^−3^)	2.45 × 10^−4^ ± 2.52 × 10^−4^ (6.11 × 10^−5^–9.94 × 10^−4^)	1.17 × 10^−2^ ± 1.20 × 10^−2^ (2.91 × 10^−3^–4.73 × 10^−2^)	7.00 × 10^−2^ ± 1.82 × 10^−2^ (5.50 × 10^−2^–1.15 × 10^−1^)
Brazil Nuts	1.14 × 10^−3^ ± 3.58 × 10^−4^ (8.89 × 10^−4^–1.90 × 10^−3^)	1.71 × 10^−6^ ± 5.37 × 10^−7^ (1.33 × 10^−6^–2.86 × 10^−6^)	5.42 × 10^−2^ ± 1.71 × 10^−2^ (4.23 × 10^−2^–9.07 × 10^−2^)	6.66 × 10^−5^ ± 6.01 × 10^−5^ (1.52 × 10^−5^–2.16 × 10^−4^)	4.20 × 10^−7^ ± 3.79 × 10^−7^ (9.58 × 10^−8^–1.36 × 10^−6^)	9.52 × 10^−4^ ± 8.59 × 10^−4^ (2.17 × 10^−4^–3.09 × 10^−3^)	3.45 × 10^−3^ ± 5.40 × 10^−4^ (2.65 × 10^−3^–4.49 × 10^−3^)	2.93 × 10^−8^ ± 4.59 × 10^−9^ (2.25 × 10^−8^–3.82 × 10^−8^)	1.41 × 10^−2^ ± 2.20 × 10^−3^ (1.08 × 10^−2^–1.83 × 10^−2^)	1.25 × 10^−4^ ± 1.62 × 10^−4^ (4.07 × 10^−5^–6.13 × 10^−4^)	5.94 × 10^−3^ ± 7.69 × 10^−3^ (1.94 × 10^−3^–2.92 × 10^−2^)	7.52 × 10^−2^ ± 1.54 × 10^−2^ (6.29 × 10^−2^–1.07 × 10^−1^)
Cashews	1.41 × 10^−3^ ± 7.69 × 10^−4^ (5.66 × 10^−4^–3.00 × 10^−3^)	2.12 × 10^−6^ ± 1.15 × 10^−6^ (8.49 × 10^−7^–4.50 × 10^−6^)	6.73 × 10^−2^ ± 3.66 × 10^−2^ (2.70 × 10^−2^–1.43 × 10^−1^)	5.57 × 10^−4^ ± 1.20 × 10^−3^ (4.41 × 10^−5^–4.16 × 10^−3^)	3.51 × 10^−6^ ± 7.54 × 10^−6^ (2.78 × 10^−7^–2.62 × 10^−5^)	7.95 × 10^−3^ ± 1.71 × 10^−2^ (6.30 × 10^−4^–5.95 × 10^−2^)	8.42 × 10^−4^ ± 7.61 × 10^−4^ (1.03 × 10^−4^–2.27 × 10^−3^)	7.16 × 10^−9^ ± 6.47 × 10^−9^ (8.72 × 10^−10^–1.93 × 10^−8^)	3.44 × 10^−3^ ± 3.11 × 10^−3^ (4.19 × 10^−4^–9.26 × 10^−3^)	1.91 × 10^−4^ ± 1.41 × 10^−4^ (4.68 × 10^−5^–4.71 × 10^−4^)	9.08 × 10^−3^ ± 6.72 × 10^−3^ (2.23 × 10^−3^–2.24 × 10^−2^)	8.77 × 10^−2^ ± 4.32 × 10^−2^ (5.16 × 10^−2^–1.64 × 10^−1^)
Hazelnuts	1.24 × 10^−3^ ± 6.17 × 10^−4^ (8.96 × 10^−4^–3.03 × 10^−3^)	1.86 × 10^−6^ ± 9.25 × 10^−7^ (1.34 × 10^−6^–4.55 × 10^−6^)	5.90 × 10^−2^ ± 2.94 × 10^−2^ (4.27 × 10^−2^–1.44 × 10^−1^)	4.95 × 10^−4^ ± 2.91 × 10^−4^ (1.44 × 10^−4^–9.75 × 10^−4^)	3.12 × 10^−6^ ± 1.83 × 10^−6^ (9.08 × 10^−7^–6.14 × 10^−6^)	7.07 × 10^−3^ ± 4.16 × 10^−3^ (2.06 × 10^−3^–1.39 × 10^−2^)	3.18 × 10^−3^± 3.96 × 10^−4^ (2.71 × 10^−3^–3.99 × 10^−3^)	2.70 × 10^−8^ ± 3.36 × 10^−9^ (2.30 × 10^−8^–3.39 × 10^−8^)	1.30 × 10^−2^ ± 1.61 × 10^−3^ (1.11 × 10^−2^–1.63 × 10^−2^)	1.90 × 10^−4^ ± 1.97 × 10^−4^ (5.74 × 10^−5^–7.72 × 10^−4^)	9.05 × 10^−3^ ± 9.37 × 10^−3^ (2.73 × 10^−3^–3.68 × 10^−2^)	8.81 × 10^−2^ ± 2.88 × 10^−2^ (6.62 × 10^−2^–1.64 × 10^−1^)
Macadamia Nuts	1.62 × 10^−3^ ± 1.37 × 10^−3^ (8.31 × 10^−4^–5.09 × 10^−3^)	2.42 × 10^−6^ ± 2.06 × 10^−6^ (1.25 × 10^−6^–7.64 × 10^−6^)	7.69 × 10^−2^ ± 6.54 × 10^−2^ (3.96 × 10^−2^–2.43 × 10^−1^)	2.91 × 10^−5^ ± 2.40 × 10^−5^ (3.86 × 10^−6^–7.85 × 10^−5^)	1.83 × 10^−7^ ± 1.51 × 10^−7^ (2.43 × 10^−8^–4.94 × 10^−7^)	4.16 × 10^−4^ ± 3.42 × 10^−4^ (5.52 × 10^−5^–1.12 × 10^−3^)	1.84 × 10^−2^ ± 3.99 × 10^−2^ (7.19 × 10^−5^–1.08 × 10^−1^)	1.56 × 10^−7^ ± 3.40 × 10^−7^ (6.11 × 10^−10^–9.22 × 10^−7^)	7.49 × 10^−2^ ± 1.63 × 10^−1^ (2.93 × 10^−4^–4.43 × 10^−1^)	1.01 × 10^−4^ ± 5.91 × 10^−5^ (4.48 × 10^−5^–2.30 × 10^−4^)	4.80 × 10^−3^ ± 2.81 × 10^−3^ (2.13 × 10^−3^–1.09 × 10^−2^)	1.57 × 10^−1^ ± 2.03 × 10^−1^ (4.72 × 10^−2^–6.89 × 10^−1^)
Peanuts	1.51 × 10^−3^ ± 7.97 × 10^−4^ (8.86 × 10^−4^–2.99 × 10^−3^)	2.27 × 10^−6^ ± 1.20 × 10^−6^ (1.33 × 10^−6^–4.49 × 10^−6^)	7.21 × 10^−2^ ± 3.80 × 10^−2^ (4.22 × 10^−2^–1.42 × 10^−1^)	3.55 × 10^−3^ ± 2.91 × 10^−3^ (1.62 × 10^−3^–1.23 × 10^−2^)	2.24 × 10^−5^ ± 1.83 × 10^−5^ (1.02 × 10^−5^–7.73 × 10^−5^)	5.07 × 10^−2^ ± 4.16 × 10^−2^ (2.31 × 10^−2^–1.75 × 10^−1^)	7.92 × 10^−3^ ± 1.54 × 10^−2^ (2.80 × 10^−3^–5.69 × 10^−2^)	6.73 × 10^−8^ ± 1.31 × 10^−7^ (2.38 × 10^−8^–4.83 × 10^−7^)	3.23 × 10^−2^ ± 6.29 × 10^−2^ (1.14 × 10^−2^–2.32 × 10^−1^)	1.14 × 10^−4^ ± 4.45 × 10^−5^ (4.66 × 10^−5^–1.92 × 10^−4^)	5.43 × 10^−3^ ± 2.12 × 10^−3^ (2.22 × 10^−3^–9.14 × 10^−3^)	1.61 × 10^−1^ ± 1.03 × 10^−1^ (8.75 × 10^−2^–4.60 × 10^−1^)
Pecans	6.05 × 10^−3^ ± 4.48 × 10^−3^ (9.91 × 10^−4^–1.32 × 10^−2^)	9.08 × 10^−6^ ± 6.72 × 10^−6^ (1.49 × 10^−6^–1.98 × 10^−5^)	2.88 × 10^−1^ ± 2.13 × 10^−1^ (4.72 × 10^−2^–6.29 × 10^−1^)	3.13 × 10^−3^ ± 1.50 × 10^−3^ (1.14 × 10^−3^–5.17 × 10^−3^)	1.97 × 10^−5^ ± 9.44 × 10^−6^ (7.17 × 10^−6^–3.26 × 10^−5^)	4.47 × 10^−2^ ± 2.14 × 10^−2^ (1.63 × 10^−2^–7.38 × 10^−2^)	4.12 × 10^−2^ ± 5.98 × 10^−2^ (5.05 × 10^−4^–1.33 × 10^−1^)	3.50 × 10^−7^ ± 5.08 × 10^−7^ (4.29 × 10^−9^–1.13 × 10^−6^)	1.68 × 10^−1^ ± 2.44 × 10^−1^ (2.06 × 10^−3^–5.43 × 10^−1^)	4.89 × 10^−4^ ± 5.66 × 10^−4^ (8.56 × 10^−5^–1.75 × 10^−3^)	2.33 × 10^−2^ ± 2.70 × 10^−2^ (4.07 × 10^−3^–8.34 × 10^−2^)	5.24 × 10^−1^ ± 3.21 × 10^−1^ (1.07 × 10^−1^–1.19 × 10^0^)
Pine Nuts	2.26 × 10^−3^ ± 1.57 × 10^−3^ (9.29 × 10^−4^–5.94 × 10^−3^)	3.39 × 10^−6^ ± 2.36 × 10^−6^ (1.39 × 10^−6^–8.90 × 10^−6^)	1.08 × 10^−1^ ± 7.49 × 10^−2^ (4.43 × 10^−2^–2.83 × 10^−1^)	1.04 × 10^−2^ ± 7.24 × 10^−3^ (1.50 × 10^−3^–1.92 × 10^−2^)	6.53 × 10^−5^ ± 4.56 × 10^−5^ (9.45 × 10^−6^–1.21 × 10^−4^)	1.48 × 10^−1^ ± 1.03 × 10^−1^ (2.14 × 10^−2^–2.75 × 10^−1^)	2.05 × 10^−3^ ± 2.97 × 10^−3^ (5.01 × 10^−4^–1.12 × 10^−2^)	1.74 × 10^−8^ ± 2.52 × 10^−8^ (4.26 × 10^−9^–9.51 × 10^−8^)	8.37 × 10^−3^ ± 1.21 × 10^−2^ (2.04 × 10^−3^–4.57 × 10^−2^)	1.67 × 10^−4^ ± 8.26 × 10^−5^ (9.48 × 10^−6^–2.81 × 10^−4^)	7.93 × 10^−3^ ± 3.94 × 10^−3^ (4.51 × 10^−4^–1.34 × 10^−2^)	2.72 × 10^−1^ ± 1.60 × 10^−1^ (1.05 × 10^−1^–5.54 × 10^−1^)
Pistachio Nuts	7.93 × 10^−3^ ± 2.28 × 10^−3^ (3.90 × 10^−3^ -1.18 × 10^−2^)	1.19 × 10^−5^ ± 3.43 × 10^−6^ (5.85 × 10^−6^–1.77 × 10^−5^)	3.78 × 10^−1^ ± 1.09 × 10^−1^ (1.86 × 10^−1^–5.62 × 10^−1^)	1.88 × 10^−4^ ± 1.63 × 10^−4^ (2.60 × 10^−5^–5.57 × 10^−4^)	1.19 × 10^−6^ ± 1.03 × 10^−6^ (1.64 × 10^−7^–3.51 × 10^−6^)	2.69 × 10^−3^ ± 2.33 × 10^−3^ (3.71 × 10^−4^–7.96 × 10^−3^)	9.13 × 10^−4^ ± 8.54 × 10^−4^ (1.36 × 10^−4^–3.27 × 10^−3^)	7.76 × 10^−9^ ± 7.26 × 10^−9^ (1.16 × 10^−9^–2.78 × 10^−8^)	3.73 × 10^−3^± 3.49 × 10^−3^ (5.56 × 10^−4^–1.34 × 10^−2^)	1.61 × 10^−4^± 1.19 × 10^−4^ (5.50 × 10^−5^–4.18 × 10^−4^)	7.65 × 10^−3^ ± 5.65 × 10^−3^ (2.62 × 10^−3^–1.99 × 10^−2^)	3.92 × 10^−1^ ± 1.09 × 10^−1^ (1.93 × 10^−1^–5.73 × 10^−1^)
Walnuts	1.73 × 10^−3^ ± 5.46 × 10^−4^ (8.09 × 10^−4^–2.71 × 10^−3^)	2.59 × 10^−6^ ± 8.18 × 10^−7^ (1.21 × 10^−6^–4.07 × 10^−6^)	8.22 × 10^−2^ ± 2.60 × 10^−2^(3.85 × 10^−2^–1.29 × 10^−1^)	2.30 × 10^−4^ ± 4.22 × 10^−4^ (6.43 × 10^−6^–1.41 × 10^−3^)	1.45 × 10^−6^ ± 2.66 × 10^−6^ (4.05 × 10^−8^–8.90 × 10^−6^)	3.29 × 10^−3^ ± 6.03 × 10^−3^ (9.18 × 10^−5^–2.02 × 10^−2^)	5.49 × 10^−4^ ± 2.52 × 10^−4^ (2.39 × 10^−4^–9.35 × 10^−4^)	4.67 × 10^−9^ ± 2.14 × 10^−9^ (2.03 × 10^−9^–7.95 × 10^−9^)	2.24 × 10^−3^± 1.03 × 10^−3^ (9.76 × 10^−4^–3.82 × 10^−3^)	3.16 × 10^−4^± 5.38 × 10^−4^ (3.97 × 10^−5^–1.88 × 10^−3^)	1.50 × 10^−2^ ± 2.56 × 10^−2^ (1.89 × 10^−3^–8.95 × 10^−2^)	1.03 × 10^−1^ ± 3.60 × 10^−2^ (4.65 × 10^−2^–1.83 × 10^−1^)

CR—cancer risk; EDI—estimated daily intake; HI—hazard index; THQ—target hazard quotient.

## Data Availability

Data are available from the authors.

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
