# Peer review of "Assessment of the Safe Consumption of Nuts in Terms of the Content of Toxic Elements with Chemometric Analysis"

_nutrients, 2021, doi:10.3390/nu13103606_

Round 1
Reviewer 1 Report
The contents of the manuscript are out of date with the current state of analytical science.

Author Response
Dear Reviewer 1,
We would like to thank you for careful and thorough reading of our paper and for the thoughtful comments and constructive suggestions, which improve the quality of the manuscript. We appreciate the time and effort that you dedicated to providing feedback on our manuscript.
We have incorporated your suggestions. Please see below for a point-by-point response to your comments and concerns.
I have a serious scientific objection to the manuscript when it keeps talking about concentrations of arsenic, mercury, cadmium and lead. It does mention in the introduction that compound forms are important, but this is not reflected in the experimental results. This is a professional mistake! We cannot talk about the toxicity and health hazards of arsenic in such general terms. It is very important to know what form of the compound it enters the body. It is a well-known fact that inorganic arsenic compounds (As3+ and As5+ compounds) are highly toxic: 70-180 mg As2O3 - PTWI (Personal Tolerably Weekly Intake). However, organic arsenic compounds (arsenobetaine, arseno-sugars, arsenocholine) are classified as "non-toxic" or "low toxic" (methylated derivatives).
I have the same objection to mercury. The same objection applies to mercury.
Thank you very much for your suggestions. Indeed, speciation analysis is an important approach and we want to do research on this in the future. However, according to our opinion, screening tests that assess contamination in food products are necessary - to know which products to examine in detail and then to determine the forms of contamination.
Screening studies can be carried out on a large number of samples to assess the exposure of the population. Earlier publications on the speciation of impurities in nuts covered a much smaller number of samples (less than 10):
Kannamkumarath, S.S.; Wróbel, K.; Wróbel, K.; Caruso, J.A. Speciation of arsenic in different types of nuts by Ion Chromatography-Inductively Coupled Plasma Mass Spectrometry. J. Agric. Food Chem., 2004, 52, 1458-1463.
In addition, studies show that the inorganic forms of As in nuts, As (III) and As (V), constitute a much greater proportion, e.g. in almonds 74%, Brazil 70% -75%, cashew 83%.
However, I miss the analysis of selenium and its compounds, which would be of particular interest in the case of Brazil nuts.
We did not include Se as a toxic element because of its beneficial effect on the human organism - when it is consumed in amounts consistent with the demand at the level of the Estimated Average Requirement (EAR). We agree that Se may have a toxic effect when consumed in large amounts. Studies by other authors indicate that Brazil nuts contain mainly organic form: for example, selenomethionine accounts for 74% or 91% of the total Se content.
da Silva, E.G.; Mataveli, L.R.V.; Arruda, M.A.Z. Speciation analysis of selenium in plankton, Brazil nuts and human urine samples by HPLC-ICP-MS. Talanta, 2013, 110, 53-57.
Lima, L.W.; Stonehouse GC, Walters C, Mehdawi AFE, Fakra SC, Pilon-Smits EAH. Selenium Accumulation, Speciation and Localization in Brazil Nuts (Bertholletia excelsa H.B.K.). Plants (Basel). 2019 Aug 16;8(8):289.
Selenomethionine seems to be the safest chemical form of selenium and appropriate form for human nutritional selenium supplementation. It is widely used in over-the-counter dietary supplements. We missed out selenium in our paper, because it is usually considered as a micronutrient. Contrary to the toxic elements analysed in our work, it is a component of dietary supplements.
However, thank you very much for drawing attention to the aspect of selenium toxicity associated with the consumption of Brazil nuts. We will take your valuable comment into account in our next manuscript on the content of beneficial elements in nuts.
In conclusion, the contents of the manuscript are out of date with the current state of analytical science.
We hope that the current, revised version of the manuscript will receive a positive recommendation from you.
We submitted the next version of our manuscript entitled “Assessment of the safe consumption of nuts in terms of the content of toxic elements with chemometric analysis” through the online submission system. The revised parts of the paper were marked. We used the "Track Changes" function in Microsoft Word. Once again, thank you for the time you put in reviewing our paper and for your recommendation for publication.
Kind regards,
Anna Puścion-Jakubik
Reviewer 2 Report
The work (ID: nutrients-1389068) is original and meets the profile required by the "Nutrients" journal. In addition, the authors present a well-structured manuscript, a topic that has not yet been explored, well-discussed results, recent references, and statistical analysis applied to the dataset is well used.
Title:
I suggest removing "elements" from the title. Maybe to: "Assessment of the safe consumption of nuts in terms of the content of toxic elements with chemometric analysis".
Abstract:
• Define abbreviations: THQ, CR, and HI.
• It remained to explore further, in summary, the purpose of the chemometric method used, which/which were, the main results obtained with them.
Introduction:
1. Briefly insert information about the oxidation state of each element. The toxicity of the elements in inorganic form depends on the oxidation state that is then present in the ingested food.
2. In the purpose of the study, the use of chemometric methods was lacking; perhaps it can be changed to: "Furthermore, the differences in the content of studied elements regarding the type of nut as well as the country of origin were determined and discriminated by multivariate analysis.
2. Materials and Methods
2.6. Health risk assessment
Although the authors indicate the formulas for health risk assessment [ref12]. I believe that inserting them in the text would facilitate the interpretation of the work. In addition, authors can briefly describe the concept for each analysis (estimated daily intake (EDI), cancer risk (CR), target hazard quotient (THQ), and hazard index (HI).
In section: "2.7. Statistical analysis" change to: "2.7. Statistical and multivariate analysis."
Chemometric approaches were performed using the same software. If yes, specify in section 2.7.
Line 113 - It is essential to state, even if briefly, the digestion conditions, especially the temperature and power used, even in a closed digestion system, there may be losses of elements with low boiling bridges during the opening of the digestion tubes, such as the case of As.
Line 135 - It is interesting to insert which type of graphite tube was used in the atomization of the Cd. The combination tube coating + chemical modifier can interfere with the efficient atomization of the analyte. Also, why was ammonium salt used as a chemical modifier for Cd analysis? Have other modifiers been tested? Generally, it is recommended to use a transition metal cation for transition metal analysis, eg Rh, Ir, W…
3. Results and discussion
Line 119 to 206 - The paragraph was confused and difficult to understand. I suggest rewriting.
Line 193 - In addition to environmental pollution, these toxic compounds may have migrated from the packaging. It would be good to include this information in this section (of the studies with nuts or other food matrices).
Line 228 – Define PNFSS
Line 217 to 218 and 228 - It is essential to remember the LOD of the As and the limits established by legislation. I suggest putting it in parentheses. Do this for all other paragraphs where this information is cited.
Line 251 and 264 – Change "brazil" to "Brazil".
Line 299 to 305 - there was no further exploration of the results obtained by cluster analysis, for example, explaining the reason for the greater similarity between the other types of nuts that presented the smallest Euclidean distances; which variables contributed to this effect?
• Table 1 - standardize the significant digits
• Table 2 – Changing p-values for Cashews superscript
Line 298-299: Enter the applied correlation type "Spearman" or "Pearson" or which one was used.
Line 290-298: The authors can further discuss the results obtained by the correlation.
Line 300-301: Insert the value of the jump considered for the HCA analysis that made it
possible to obtain the four main clusters. I believe it is a dissimilarity close to 2.3.
• In the legend of figure 1, change "he atmap" for "heatmap".
• Figure 2 for HCA can be improved if each of the four formed clusters is shown in
different colors, which would make it easier for readers to see.
Line 312 - A biplot type graph where it is possible to visualize the scores and loadings may
be better to visualize the variables that were responsible for the discrimination of the groups
in the PCA
Round 2
Reviewer 1 Report
I accept the reasons given by the authors in the cover letter.
I also accept the additions made in the manuscript
Reviewer 2 Report
The authors answered all my questions point by point. For me, the article is ready to be published on Nutrients.